# A Multi-Point Identification Approach for the Recognition of Individual Leopards (*Panthera pardus kotiya*)

**DOI:** 10.3390/ani12050660

**Published:** 2022-03-06

**Authors:** Milinda Wattegedera, Dushyantha Silva, Chandana Sooriyabandara, Prasantha Wimaladasa, Raveendra Siriwardena, Mevan Piyasena, Ranjan M. S. L. R. P. Marasinghe, Bhagya M. Hathurusinghe, Rajapakse M. R. Nilanthi, Sadeepa Gunawardena, Heshan Peiris, Pasan Seneviratne, Pramod C. Sendanayake, Chathura Dushmantha, Sudantha Chandrasena, Sahan S. Gooneratne, Pumudi Premaratne, Sandaru Wickremaratne, Mindaka Mahela

**Affiliations:** 1The Leopard Research and Data Gathering Team, Yala Leopard Visitor Center, Yala National Park, Palatupana 82614, Sri Lanka; milindawattegedara@gmail.com (M.W.); prasanwimal2020@gmail.com (P.W.); raveendras74@gmail.com (R.S.); mevan30@gmail.com (M.P.); 2Department of Wildlife Conservation, 811A Jayanthipura, Battaramulla 10120, Sri Lanka; csooriyabandara@gmail.com (C.S.); ranjanm.bpm@gmail.com (R.M.S.L.R.P.M.); nilanthi.dwc@gmail.com (R.M.R.N.); 3Post Graduate Institute of Science, University of Peradeniya, Peradeniya 20400, Sri Lanka; bhagya.hathurusinghe93@gmail.com; 4246/A/4, Rocky Hill Residencies, Hokandara South, Hokandara 10230, Sri Lanka; sadeepag@gmail.com (S.G.); heshpeiris4@gmail.com (H.P.); pasan4u@gmail.com (P.S.); sendayt@gmail.com (P.C.S.); dushmantha.1989@gmail.com (C.D.); dschandrasena@gmail.com (S.C.); sahan.gooneratne@yahoo.com (S.S.G.); ppumudi@gmail.com (P.P.); sandarupabasara8@gmail.com (S.W.); mcmindaka@gmail.com (M.M.)

**Keywords:** *Panthera pardus kotiya*, identification, Yala National Park, spot, rosette, population

## Abstract

**Simple Summary:**

All of the previous research on photography-based leopard identification was conducted based on the assumption that leopard spots and rosette formations do not change in shape or form. We observed 29 instances of changes to spot and rosette formations in continuously observed leopards at Yala National Park, Block 1. Since the previous approaches have flaws and errors, the same leopard may be misdiagnosed and counted numerous times, overestimating leopard populations if the spot and rosette formation of a leopard has changed. To address this issue, we developed the multi-point leopard identification method, which is a novel process for identifying Sri Lankan leopards. The minimum leopard population of Yala National Park, Block 1, on 31 March 2021, was established during the study.

**Abstract:**

Visual leopard identifications performed with camera traps using the capture–recapture method only consider areas of the skin that are visible to the equipment. The method presented here considered the spot or rosette formations of either the two flanks or the face, and the captured images were then compared and matched with available photographs. Leopards were classified as new individuals if no matches were found in the existing set of photos. It was previously assumed that an individual leopard’s spot or rosette pattern would not change. We established that the spot and rosette patterns change over time and that these changes are the result of injuries in certain cases. When compared to the original patterns, the number of spots may be lost or reduced, and some spots or patterns may change in terms of their prominence, shape, and size. We called these changes “obliterate changes” and “rejig changes”, respectively. The implementation of an earlier method resulted in a duplication of leopard counts, achieving an error rate of more than 15% in the population at Yala National Park. The same leopard could be misidentified and counted multiple times, causing overestimated populations. To address this issue, we created a new two-step methodology for identifying Sri Lankan leopards. The multi-point identification method requires the evaluation of at least 9–10 spot areas before a leopard can be identified. Moreover, the minimum leopard population at the YNP 1 comprises at least 77 leopards and has a density of 0.5461 leopards per km^2^.

## 1. Introduction

Much research has relied on the detection of variation in nature to recognize individual animals [1]. In the Felidae, the patterns that are displayed on the flank and face are highly variable. These spots, rosettes, or strips are distinct to each individual [2]. Previous studies on the adaptive function of coat patterns have indicated that they are more likely for camouflage rather than for communication or physiological reasons [3]. The spots in the Felidae are significantly associated with arboreality [4]. McDougal, in 1977, used tigers (*Panthera tigris*) variable markings on either side of the face, legs, and shoulders for individual identification. Moreover, the individual identification of leopards (*Panthera pardus*) has been accomplished by using spot pattern markings [5] or spot pattern variation [6]. Some researchers have emphasized that spot patterns vary among individual leopards according to the side of the muzzle, the number of spots, and their position relative to each other as well as bilaterally within individuals and that in addition to variability in spot patterns and color, size differences indicate the sex of an animal [6]. Moreover, the pattern of the rosettes and spots is unique to each individual animal [7].

In order to assess spot pattern variation, several modeling techniques have been adopted, with the capture–recapture method remaining a popular method [8]. For sensitive, threatened species, in particular, the development of non-invasive recognition methods that avoid the direct handling and tagging of the study subjects is advised for procedures such as the marking–recapture method [8,9,10]. Camera-trapping, which has been practiced for a long time, is a potential tool that can be used to assess the population of elusive carnivore species in ecological sciences, as it is able to recognize individuals by their distinctive pelage patterns with minimum disruption [11,12]. Capture–recapture methods have been widely used to estimate abundance and density from camera photos in many carnivores, such as in the snow leopard (*P. uncia*), tiger [13,14,15], bobcat (*Lynx rufus*), (Alonso et al. 2015), black bear (*Ursus americanus*) (Fusaro et al. 2017), jaguar (*P. onca*) and common leopard (*P. pardus*) [5,16,17,18,19].

In the case of leopards, so far, all visual leopard identification procedures have been conducted by adopting the capture–recapture method through camera traps and DSLR photographs [11,20,21,22]. The capture–recapture method only considers the areas of the skin that are captured by the equipment. Here, the spot or the rosette formations on either the two flanks or the face are compared [6]. The captured images are then compared to the available photographs in the database and are matched. If no matches are made with the existing database, then these leopards were identified as new individuals [8]. However, during capture–recapture methods, marks are considered to be stable for the duration of the study. Therefore, misidentifications can occur, and estimates must account for marks that may change over time. 

During the capture–recapture method, which is practiced internationally, the same leopard can be misidentified and counted many different times due to the inherent weaknesses of the method. This results in an overestimation of leopard populations because of the duplication of identities. 

Our study species was the Sri Lankan leopard (*P. p. kotiya*), which is an endangered endemic subspecies of Felidae. It lives in montane, sub-montane, tropical rain, monsoonal dry evergreen, and arid zone scrub forests [23]. It is an apex predator and the largest of Sri Lanka’s four wild cat species. It was first described in 1956 by P.E.P. Deraniyagala [5,24,25,26,27,28]. It has a rusty yellow coat with dark spots and rosettes without central spots with individually unique patterns, which are smaller than in Indian leopards [29]. Melanistic individuals have also been reported [19,30].

Therefore, to overcome the above-mentioned challenges and weaknesses of the existing methods for the identification of individual leopards, we aimed to develop a method for leopard identification that does not result in a duplicated count during surveys. With the assumption that leopards change their spots over time due to observed or unobserved injury, we monitored a set of selected leopards for a longer period of time. We also observed a wide area of the facial and flank area of leopards. Our aim was to develop a non-invasive, quick, and accurate method that could be employed for Sri Lankan leopards. 

## 2. Materials and Methods

### 2.1. Study Area

The study was conducted in Yala National Park, Block I, the same area in which Kittle et al. carried out their study [5]. The Yala National Park, Block 1 (YNP1) [31] is an area that is protected by the Department of Wildlife Conservation of Sri Lanka. The park is located in the Southern province and extends to the Uva province of Sri Lanka. Yala National Park, Block 1, spans a total area of 141,000 hectares. The research area was determined using a GIS unit belonging to the Department of Wildlife Conservation. 

### 2.2. Methodology

Data were obtained from 750 days of direct observation and over 2250 sightings during the period from 2012 to 2021. Travel inside YNP1 was only carried out in vehicles. The leopards were photographed using Nikon, Canon, Fuji, and single-lens reflex (DSLR) cameras. Images were recorded in digital image files in the RAW or JPG formats in the DSLR cameras, and various ranges of telephoto and fixed lenses were used to capture the images. The images from the SLR cameras were captured on Kodak and Fuji film, and the negatives of these images were stored on compact disks. All of the images are stored in digital storage devices in the same format as the original Photo (Figure 1).

### 2.3. The Visual Morphological Description of Sri Lankan Leopard 

The shapes, sizes, and formations of spot and rosette formations are unique to every leopard, but the positioning of such spot and rosette formations in each leopard’s skin coat were common. 

### 2.4. General Markings

The leopard’s striking rust yellow coloration and blackish markings make it easy to identify. The size, appearance, and shape of the spots vary within the coats of individual leopards (Figure 2A). The individual spots form a pattern in some parts of the coat (Figure 2A). In some areas of the coat, individual spots form an amalgamation and create a rosette (Figure 2A). In some areas of the coat, a darker orangish contrast can be observed between the spots that form a rosette. Individual rosettes in the skin will form into a collective pattern (Figure 2A). The shape of such a pattern is subjective to each observer. Rosettes can be of different sizes and shapes. Leopard skin has markings that are unique to individual leopards. We have not come across any leopards bearing the same spot and rosette pattern formations. There are individual spots that appear similar when visually examined. 

The color of a leopard’s skin coat between the blackish spot and rosette formations is rust yellow on the sides of the body, face, upper area of the leopard’s body, and tail, as shown in Figure 2B(a). The color of the coat underneath the leopard’s body and on the inner sides of the legs of the leopard is white or a mix of white and off-white. The whitish part, shown in Figure 2B(b), only consists of spot markings, and no rosette markings are present.

Leopards have distinctive markings on the bottom of their abdomen (belly area) that consist of a formation of thick spots in the shape of a circular or oblong circumferential pattern, with fewer spots within the perimeter. This pattern was found in all of the leopards we examined. The spot patterns and formations of individual leopards, on the other hand, vary, though the presence of such a pattern is common. However, the spot patterns and formations are different in individual leopards, though the presence of such a pattern is common. We have termed this as Idiosyncratic spot pattern formation. A comparison of the idiosyncratic spot pattern formation of three individual leopards that were identified is shown in Figure 2C(a–c). 

### 2.5. Identification of a Leopard 

The identification of leopards in the wild is performed by observing the spot and rosette patterns. The obliterate and rejig spot changes were not considered previously. This change creates limitations in accurate identification of a leopard. 

When we were observing the spot pattern variation within the leopards in the area, and when we were taking the population count, we encountered some instances, which we were misled during the capture–recapture method. Therefore, we further studied these misidentifications and employed a step forward in capture–recapture method. 

In considering these limitations of identifying a leopard, we have established that multiple areas of the coat of a leopard should be considered when identifying a leopard. These multiple areas of a leopard’s coat have been segregated in our model. Our research has established a new two-step methodology in leopard identification termed by us as the multi-point leopard identification method. 

We have segregated the leopard’s coat to fifteen (15) points according to the spot and rosette formations of those areas for the purpose of identification. Each segregation is termed as a point. The points have been enumerated, and the number is termed as the multi-point reference code (MPC points). This segregation was performed for the identification of Leopards to be highly accurate. We suggest that the spot and rosette formations of a minimum 9 MPC points out of the 15 MPC points must be thoroughly compared with existing images before classifying a leopard as a newly identified leopard and when establishing the identity of an already identified leopard. The accuracy of the identification will increase through the comparison of more MPC points. The segregation of the leopard’s skin coat and the MPC Points that are used in the multi-point leopard identification method is shown below in Table 1.

### 2.6. Coding of Identified Leopards

Each identified leopard was provided with a sex-based identification code. Even though a particular leopard was identified before this age, the code was only meant to be assigned to leopards beyond the age of 1.5 years. For simplicity of use, the code can be abbreviated and consists of a sex-based alphabetical abbreviation and a numeric value. For each sex, the numerical value is continuous. For easier identification, each coded leopard is also assigned a nickname. Once the code and nickname have been assigned, they are added to our leopard database. Observation records of the relevant leopard are kept in the database and are updated after each observation.

### 2.7. Data Collection for Minimum Population Estimation of Leopards at YNP1

We conducted a census in order to determine the current population of the leopards that were recognized during the identification period. By 31 March 2021, the census had been completed. The leopards that we had identified and listed after the survey was completed (leopards older than 1.5 years) were not considered for the results, except for in population and density calculations. A survey of the sightings of individually identified leopards was used to conduct the census. The survey was conducted using a census survey form in which participants were required to name the month and year of their last photographed leopard. The census survey form included the leopard’s identification code and nickname as well as a cage to indicate the month and year in which the specific leopard was last photographed by the participant with a DSLR camera. The identified leopard population was calculated using observations from January 2020 to March 2021, a 15-month period. A 15-month period was explored to compensate for sightings that were lost due to the park’s closure for three months in 2020 due to the COVID-19 pandemic. The participants were selected by us and by the DWC. The participants were selected using an expert-stratified nonprobability sampling method, basing their knowledge on the identified leopards. All of the participants were conversant with the leopard identification codes and names through the researcher’s publication “Leopard Diary, Yala National Park, Block-01, Volume 1”, which was published under ISBN 978-624-96411-0-5 [31] and is currently in use as the guidebook for in-field leopard identification by the staff of the DWC and other leopard photographers.

Participants (excluding DWC employees) were required to produce the original photograph of their last seen leopard photograph for verification. The DWC participants certified that they saw the individual leopards during the time duration mentioned. The results of all 25 participants’ sightings were amalgamated and analyzed. The leopards were categorized according to their estimated age (Table 2). 

Age estimation was achieved by considering the observation period of a particular leopard when photographic data was available. If no data were available, we conducted a detailed comparison of the leopard’s physical appearance with those of the many leopards in our database. The unlisted but observed leopards as of 31 March 2021 were considered for the calculation of the minimum leopard population of the YNP for the study period. These leopards were unlisted as of 31 March 2021 due to not being of the listing age and due to other criteria decided by the team. However, photographic evidence of all of the unlisted leopards is available and is currently listed (except for data of the leopards who do not qualify for the minimum listing-age criteria). 

Data were collected according to the criteria developed based on variations in the spot and rosette formations on facial area, foreleg area, and flank (Table 3). The areas used for principal component analysis are stated as PCA in the spot location column of Table 3.

### 2.8. Data Analysis 

Principal component analysis (PCA) was carried out on 19 spot distribution characteristics in order to form correlations between the different attributes under study via group clustering. PCA separates correlated attributes from uncorrelated ones. This was performed to clarify the significance of spot counting and rankings on leopard identifications. The total number of spots counted in 19 areas of 126 leopards was considered during the analysis. Moreover, cluster analysis was used to group the leopards based on spot distribution. PCA and cluster analysis was carried out using JMP v16 [32]. Based on the eigen values, which are higher than 1, the main components/characteristics that describe the clustering will be analyzed. 

By using the Euclidean distance method, Ward’s linkage cluster analysis was performed to identify pattern variation among leopard spots. We analyzed the data using DISTANCE v.7 [33] software and computed the density. 

## 3. Results

### 3.1. The Identified Leopards of Yala

The team has identified 137 individual Leopards at YNP1 since 2013 using the multi-point identification method. All the images of the leopards that had been identified before the development of the multi-point identification method were reconfirmed using the new method. A sex-based list of the leopards that have been identified in YNP1 since 2013 is shown in Table 4. There are another eight unlisted leopards, but these comprise observed Sub Adults and Cubs in YNP1 as of 24 September 2021. 

The minimum leopard population at YNP1 and the minimum leopard density at YNP 1 as of 31 March 2021 were determined. The minimum leopard population at YNP1 was determined via a cross-sectional census survey of sightings of identified individual leopards. The minimum population at YNP 1 as of 31 March 2021 was 77 and is shown in Table 4.

According to the principal component analysis, the first eight principal components have eigenvalues greater than one (Appendix A). Together, these eight components only explain 69.9% of the variation in the data. Since only a small amount of data can be explained by all of the right components, it is hard to say whether there is a correlation between each multi-point location and each individual (Figure 3). Therefore, each and every spot area is important for identifying an individual, and at least nine spot areas—F1, F2, F3, RM2, LM2, LM1, RM1, LNS, and RNS—should be considered. 

The dendrogram (Appendix A) presents the distribution of three main clusters. Cluster 1 consists of leopards with spots below the mystacial area. Cluster 2 and 3 are distinctively separated by sex. Cluster 2 consists of female leopards. Cluster 3 consists of male leopards. The significant difference among the male and female leopards is that male leopards have more spots on their left and right foreleg ranks. 

There have been no reported observations of the central spots that are commonly seen in jaguars in *Sri Lankan leopards* (Figure 4A,B). However, we noticed that leopards YM 33 (Figure 4C,D) and YM 43 (Figure 4E,F) had larger rosettes and spots in the mystacial, nasal, eye, and foreleg areas. This is a rare occurrence, as most other leopards have no central spots in either of the rosettes.

### 3.2. Changes in Spot Markings and Rosette Patterns of a Leopard

Before the start of our study, it was an accepted fact that the spot markings and rosette patterns of a leopard do not change over time. However, during our study, we observed that changes could occur in the spot markings and rosette patterns of leopards over time. As per our findings, these changes occur through injuries sustained by the leopards but may also occur without the presence of an injury. Out of 137 leopards, we continuously monitored 59 leopards and observed 29 instances of injuries. We closely observed the injuries of those identified leopards and compared them with documented photos taken of those leopards prior to such injury, documented the leopards during the injury, and continuously observed the leopard after the injury. We have presented 29 instances, and in 16 leopards, we observed changes in the spot markings and rosette patterns. Therefore, these 29 instances had previously been misidentified, and these leopards were documented as a total of 68 separate leopards, representing an error rate of 15.25%.

Changes in the spots and rosettes were observed to take two forms (Appendix A). When obliterate changes take place, a spot or rosette pattern change that was originally present is subsequently not visible. The spots are succeeded by the yellowish skin coat in the areas where obliterate changes have occurred. The rust yellow color of the skin coat may be lighter in places where obliterate changes have occurred. A rejig change is when the prominence, shape, size, and appearance of the original spot or the rosette pattern changes. 

During our study of the 59 leopards, the majority of the spot alterations were identified in the forehead areas, mystacial area, nasal spots, and left and right sides of the face (Table 5). 

### 3.3. Changes to Spot Patterns with an Injury Being Documented

Our spot and rosette pattern change observations have been presented in individual tables related to individual injuries. We observed 13 instances in 10 leopards over the 10 years of observations. Six instances from the above are documented in Figure 5, Appendix A. Figure 5 demonstrates the changes in spot markings and rosette patterns of a leopard in reference photos and the same set of photos (zoomed in for some photos) marked with red on the skin coat to demarcate the point of concern, injury, or change in the relevant photo.

### 3.4. Changes to Spot Patterns without a Documented Injury

The injuries that a wild leopard may sustain cannot always be documented. We observed 16 changes in the spot and rosette patterns of 12 leopards that took place without the documentation of a previous injury. Figure 6, Appendix A, documents eight instances of change from six leopards.

Accordingly, most of these changes were observed in the forehead area (F1,F2,F3), mystacial areas M1 and M2, and near the nasal area, near-eye, and flank areas. These changes were observed on both the right and left sides of the face and flank. Therefore, according to our observations, out of 15 points, at least these 10 points should be included and considered when identifying an individual when using the capture–recapture method. Specifically, forehead spot patterns must be included because changes can happen more often in these areas and can mislead observers. 

Moreover, during the observations, some leopards were misidentified based on sex. This is discussed in Appendix A.

## 4. Discussion

Variations in natural markings have been valuable to field biologists for the identification of individual animals. It is well recognized that photographs may be misclassified when identifying individuals using natural marks or patterns due to poor photographic quality if the variability in the marking patterns is small or if the patterns vary over time [1,34]. Inaccuracies and errors in leopard identification will result in duplicated leopard identities, resulting in the overestimation of leopard populations [1,35].

We widened the factors that need to be applied for the investigation of individual leopards. We observed the skin coat of the leopard and segregated the different markings that will be useful for future studies. We segregated the forehead markings and provided them with new terms. The first and second areas of the guide arch on either side of the forehead were called the elongated spot and the genesis spot, respectively, and these terms were determined based on the importance of these two areas. The spot formation between the elongated spots of the guide arches on either side was called F1 spot formation. We observed that F1 spot formation has changed from its original formation due to obliterate or rejig changes. F2 spot formation is the prominent marking above F1 spot formation. F2 spot formation always starts and ends inwardly parallel to the genesis spots of the guide arches on either side. F2 spot formation curves downwards towards F1 spot formation.

Studies examining spot patterns in the mystacial area, spots near the eyes, and spot patterns on the forehead have been conducted but have only covered a small number of leopards [6]. The multi-point identification method introduced in this study relies on 15 points: the skin of the leopard, covering the total face area, as well as the flank area, left and right side nasal spots, left and right side mystacial row 1, left and right side mystacial row 2, left and right side of the face, left and right side of the neck, left and right side foreleg, left and right side of flank, and the forehead (F1, F2, F3). These points should be considered when defining an individual. According to our data, it is evident that there is no correlation between the right and left sides of the face and flank, even though [6] mentioned that the only areas where vibrissae are observed other than in the mystacial area in a leopard’s face are within the mandibular spot formation and the periphery of the elongated spots. We observed central spots within the rosette pattern in some leopards. It has been stated that in mammalian species, there may be spots within the rosettes of *P. pardus*, similar to the rosettes observed on jaguars [7]. These spots inside the large rosette are referred to as central spots in a jaguar [36]. We observed pattern formation underneath the abdomen (belly) area of all of the leopards in YNP1, which consists of large spots that form a circular or oblong shape and only a few spots within the formation. We called this formation the ‘idiosyncratic pattern’. We are currently conducting further research to compare this formation among other leopard subspecies around the world.

Therefore, this newly introduced multi-point leopard identification method overcomes the inaccuracies and errors that can occur during the visual identification of leopards. In the wild, leopards are identified using capture and recapture methods, and for a long time, leopard populations were estimated by observing their markings and assuming that the spot and rosette markings do not change [15,17,37]. In 2013, Hunter mentioned that the rosette and spot patterns that are unique to individuals could be useful for identification [38]. 

The current study was carried out with the assumption that the leopard spots change over time. When we initially employed the traditional capture–recapture method, we misidentified the leopards and counted the leopards as different individuals due to spot formation changes in the face after injuries. Regardless of the extent and sophistication of the survey methods and analyses used so far based on camera-trapping data, incorrect classification creates false conclusions and biased estimates [35,39]. However, with continuous observations amounting to the recaptures of the leopards, we observed that the recognized leopard was not a new leopard but instead a leopard that we captured earlier. There was a high error rate in the misidentification rate of these leopards (~15%), and the former capture–recapture counts should be revisited. 

To overcome this issue, we employed long-term continuous monitoring and observed that this capture–recapture method may contribute to identification duplications, thereby causing the overestimation of leopard populations. Out of 137 leopards from our study, we continuously monitored 57 leopards for 10 years. Out of them, nine leopards with sustained injuries showed subsequent changes in their spot and rosette patterns. These changed spot and rosette formations did not return to their original formations on any of the leopards during photographic recaptures after such observations were made. These changes were termed “obliterate changes” and “rejig changes” based on the nature of the change.

Moreover, the main factors that help in differentiating males and females are the size, color, genitalia, and tracks/paws. Adult male leopards are larger than females [40,41]. However, we found that there was variation in the spot counts in males and females that have not been identified by previous reports. The significant difference among male and female leopards is that male leopards have a higher density of spots in left and right foreleg rank. This shows deep bifurcation between male leopards and female leopards that has not been identified by previous reports. Further studies regarding the differentiation of male and female leopards will include the full flank area. Studies examining the spot patterns in a small area have been conducted; the mystacial area, spots near eyes, and spot patterns on the forehead are only visible in a small number of leopards. No previous research has defined different leopard markings. Miththapala et al., 1989 [6] and Stein and Hayssen, 2013 [7] discuss spots and rosette patterns but do not describe them.

In one instance, we may have come to the wrong conclusion by only observing the presence and absence of testicles. Spontaneous alterations in the position of the testicles in humans have been identified in the literature [42]. The same has been found within the leopard population. Visually identifying the sex of a male leopard can be inaccurate, as we observed that the testicles of a male leopard were not visible for 10 s before subsequently resurfacing. Therefore, repeated observations need to be adopted before the sex of a leopard can be concluded.

Furthermore, for the current study, 137 leopards at Yala National Park were identified by means of DSLR photography and observation during the period from 2013 to 2021. All of the leopards were assigned a code based on gender and added to our database along with a nickname. Leopard cubs younger than 1.5 years of age were not given a code. In 2017, The estimated population density in YNP1 was 0.217 leopards per square km [5]. However, according to our records, the minimum leopard density at the YNP1 was 0.5461 leopards per square km based on the minimum population of leopards present in the YNP1 as of 31 March 2021. The increase in the leopard population density may be due to short-term changes in the environment and limited resources such as food [43,44] as well as population dynamics, which can cause variance in breeding success [39,45]. The minimum leopard population was determined by a cross-sectional census survey, and the sample participants were selected based on an expert-stratified nonprobability sampling method.

This validation could be employed not only for the leopards but also for other members of the Felidae. The quantitative measures, as well as the qualitative measures that were taken earlier, would need to be revisited. We have introduced a two-step methodology, a step forward from the traditional capture–recapture method. 

## 5. Conclusions

Our findings show that following an injury, the number of spots, as well as the spot and rosette patterns, changes. Since there is no association between spot counts in specific locations, multi-point identification must be used to identify new individual leopards. As we have shown that leopard spot patterns can change over time, we recommend at least 9–10 characteristics should be examined to identify an exact individual. 

Following the application of these described leopard survey methods, the population at this field site is estimated at YNP is 77, and the population density is 0.5461 leopards per km^2^. 

## Figures and Tables

**Figure 1 animals-12-00660-f001:**
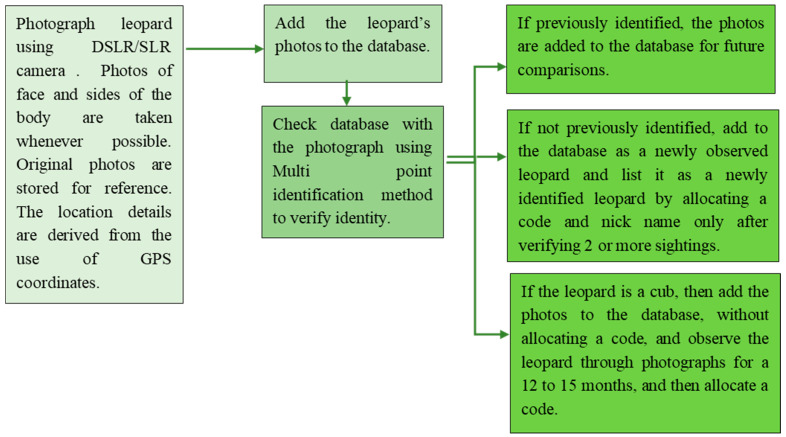
Flowchart for morphological data collection from each leopard photograph. The research was carried out with the permission and under the supervision of the Department of Wildlife. Conservation (DWC), permit No. WL/O3/02/78/15.

**Figure 2 animals-12-00660-f002:**
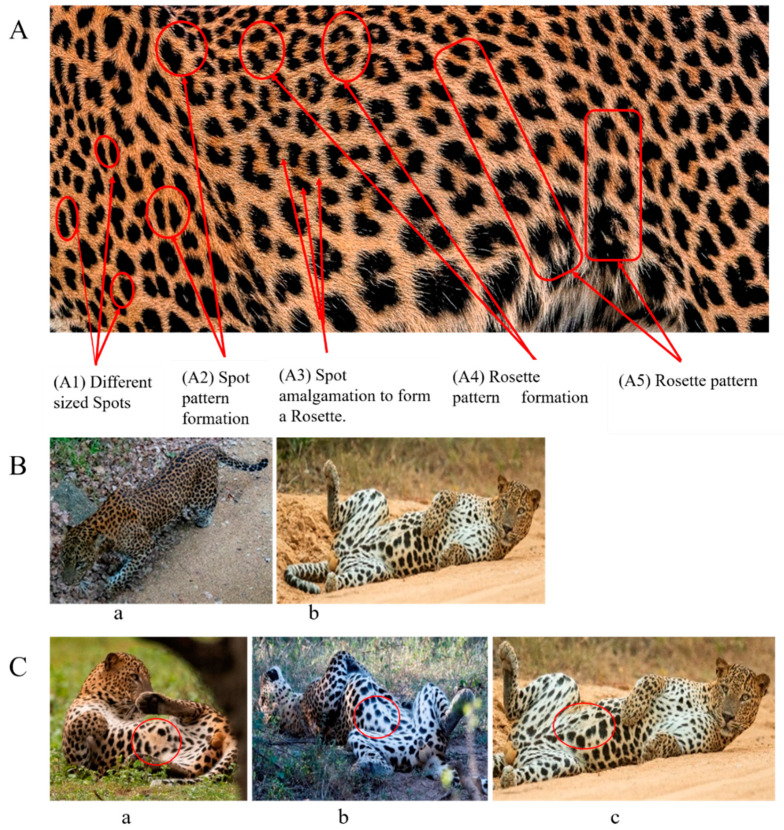
A visual morphological description of a *Sri Lankan leopard*. (**A**) The leopard’s fur coat; (**B**) color variations in the skin coat of leopards. (**a**) Top of the body; (**b**) underneath the body—photo credit, Ashvitha Wickrama; (**C**) idiosyncratic spot pattern formation. (**a**) Leopard YM 47; (**b**) Leopard YM 33; (**c**) Leopard YM 43—Photo credit, A. Wickrama.

**Figure 3 animals-12-00660-f003:**
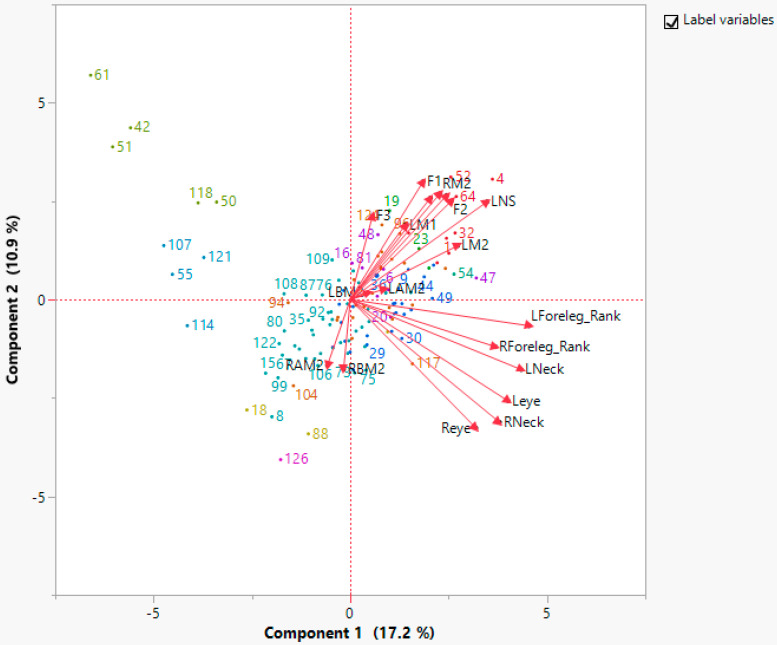
Principal component analysis (PCA) biplot of spot distribution in Yala leopards. Scatter plot of female and male leopards are represented on two major principal component axes. Variables are grouped into two principal components. PC1 represents the spots distributed on both right and left side of neck, eye, and foreleg and PC2 represent both left and right nasal and mystacial spots and spots on the forehead.

**Figure 4 animals-12-00660-f004:**
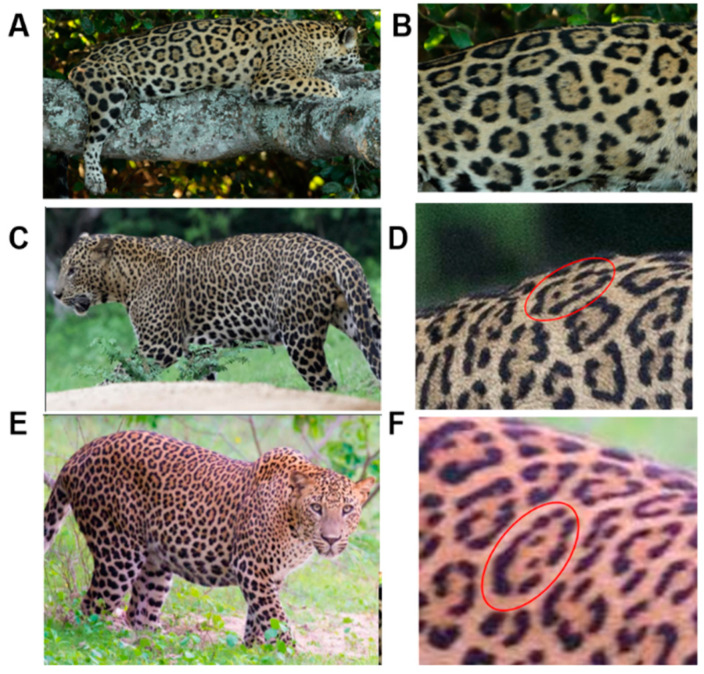
Central spots in leopard’s rosettes (**A**,**B**) *P. onca* spot formation (Photo credit. Senaka Kotagama) (**C**,**D**) YM 33 with larger rosette with spots in the middle at flank area (**E**,**F**) YM 43 with larger rosette with spots in the middle at flank area.

**Figure 5 animals-12-00660-f005:**
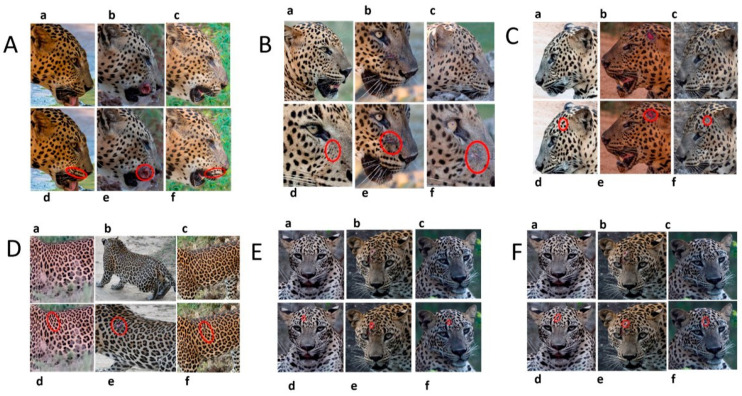
**Changes to spot patterns with an injury being documented.** Changes in spot markings and rosette patterns of a leopard in reference photos (**a**–**c**) and the same set of photos (zoomed in for some photos) marked with red (**d**–**f**) on the skin coat to demarcate the point of concern, injury, or change in the relevant photo. (**A**) **Change in the spot markings YM 07—Subsequent to Documented Injury 1 of YM 07:** (**a**–**c**) reference photos (**d**) documented original spot formation in December 2017, before injury; (**e**) documented change in spot formations in June 2018, pc—Saman Abeygunewardena; (**f**) documented spot formation changes in February 2021, pc—Pasan Seneviratne. (**B**) **Changes in the spot markings of YM 16—Subsequent to the Documented Injury 1 of YM 16:** (**a**–**c**) reference photos (**d**) documented original spot formation in July 2020; (**e**) documented spot formation changes in October 2020, pc—Gautham Kumar; (**f**) documented spot formation changes in August 2021. (**C**) **Changes in the spot markings of YM 16—Subsequent to the Documented Injury 2 of YM 16:** (**a**–**c**): reference photos; (**d**) documented original spot formation in July 2018—before injury; (**e**) documented changes in spot formations in February 2020, pc—Samith Perera; (**f**) documented changes in spot formations in September 2021. (**D**) **Change in the spot markings of YM 27—Subsequent to the Documented Injury 1 of YM 27:** (**a**–**c**) reference photos (**d**) documented original rosette formation in January 2020—before injury; (**e**) documented changes in spot formations in October 2020, pc—Janaka Dassanayake; (**f**) documented changes in rosette formations in March 2021. (**E**) **Changes in the spot markings of YM 59—Subsequent to the Documented Injury 1 of YM 59:** (**a**–**c**) reference photos (**d**) documented original spot formation in October 2020—before injury; (**e**) documented changes in spot formations in January 2021, pc—Chamli Weerasinghe; (**f**) documented change in spot formations in March 2021. (**F**) **Changes in the spot markings of YM 59—Subsequent to the Documented Injury 2 of YM 59:** (**a**–**c**) reference photos (**d**) documented original spot formation in October 2020—before injury; (**e**) documented changes in spot formations in January 2021, pc—Chamli Weerasinghe; (**f**) documented changes in spot formations in March 2021 (pc-photo credits).

**Figure 6 animals-12-00660-f006:**
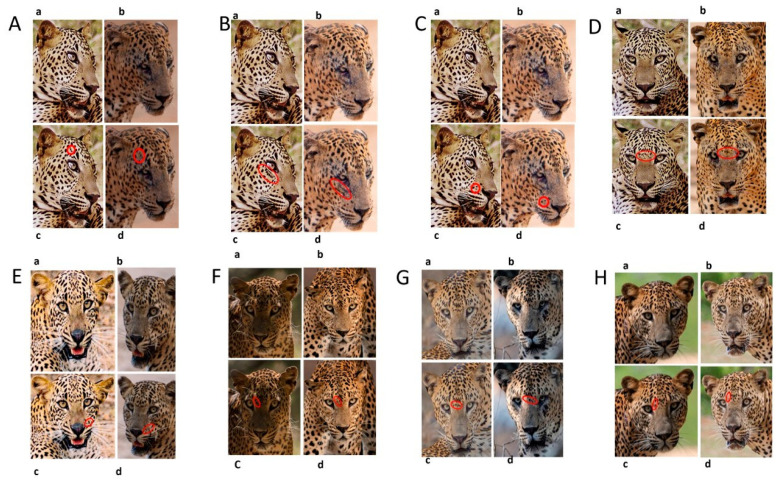
**Changes to spot patterns without a documented injury.** (**A**) **Changes in the spot markings of YM 01—Subsequent to the Undocumented cause 1 of YM 01:** (**a**,**b**) reference photos (**c**) documented original photo in July 2004; (**d**) documented changes spot formations in January 2016. (**B**) Changes in the spot markings of YM 01—Subsequent to the Undocumented cause 2 of YM 01: (**a**,**b**) reference photos; (**c**) documented original photo in July 2004; (**d**) documented changes in spot formations in January 2016. (**C**) **Changes in the spot markings of YM 01—Subsequent to the Undocumented cause 3 of YM 01:** (**a**,**b**) reference photos (**c**) documented original photo taken in July 2004; (**d**) documented changes in spot formations in January 2016. (**D**) **Changes in the spot markings of YM 01—Subsequent to the Undocumented cause 4 of YM 01**: (**a**,**b**) reference photos, pc—Nimesh Dhanasekera; (**c**) documented original photo in July 2004; (**d**) documented changes in spot formations in August 2016. (**E**) **Change in the spot markings of YF 03—Subsequent to the Undocumented cause 1 of YF 03:** (**a**,**b**) reference photos (**c**) documented original photo in 2005; (**d**) documented changes in pot formations in March 2015. (**F**) **Changes of spot markings YF 01—Subsequent to the Undocumented cause 1 of YF 01:** (**a**,**b**) reference photos; (**c**) documented original photo in April 2015; (**d**) documented changes in spot formations in August 2019. (**G**) **Changes in the spot markings of YM 07—Subsequent to the Undocumented cause 1 of YM 07:** (**a**,**b**) reference photos (**c**) documented original photo in May 2017; (**d**) documented changes in spot formations in August 2021. (**H**) **Changes in the spot markings of YM 32—Subsequent to the Undocumented cause 1 of YM 32:** (**a**,**b**) reference photos pc—Samith Perera; (**c**) documented original photo in December 2018; (**d**) documented spot changes in October 2021 (pc-photo credits).

**Table 1 animals-12-00660-t001:** Segregations of the skin coat and the multi-point reference code points used for the multi-point identification method.

Skin Coat Segregation	Left Side Nasal Spot	Left Side Mystacial Row 1	Left Side Mystacial Row 2	Left Side of the Face	Neck—Left Side	Left Side Foreleg
MPC Point	1	2	3	4	5	6
Photograph of the segregation (Marked within the red area)	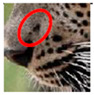	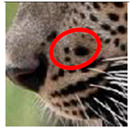	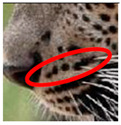	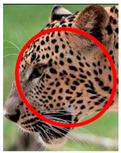	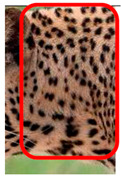	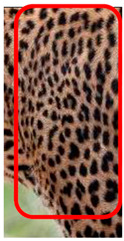
Skin Coat Segregation	Right Side Flank	Forehead	Left Side Flank
MPC Point	9	8	7
Photograph of the segregation (Marked within the red area)	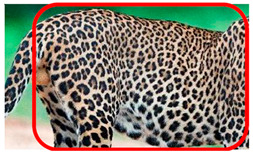	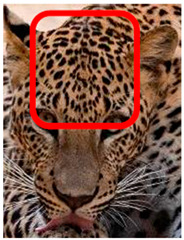	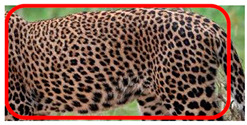
Skin Coat Segregation	Right side Foreleg	Neck—Right Side	Right Side of the Face	Right side Nasal Spots	Right Side Mystacial Row 1	Right Side Mystacial Row 2
MPC Point	10	11	12	13	14	15
Photograph of the segregation (Marked within the red area)	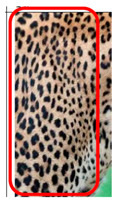	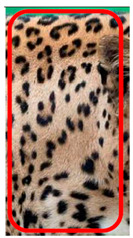	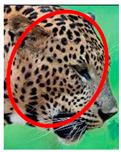	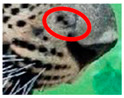	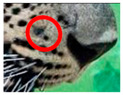	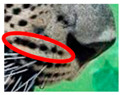

**Table 2 animals-12-00660-t002:** The basis of categorization with age.

Age Category	Abbreviation Used	Basis of Separation
Cub	Cub	A Leopard whose estimated age is 6 months.
Sub adult	SA	The period of 30 months after observation as a cub.
Early adult	EA	The period between 31 months and 45 months after observation as a cub or if the estimated age is less than 4 years.
Adult	A	An estimated age between 4 years and 8 years.
Aged adult	AA	An estimated age >8 years.

**Table 3 animals-12-00660-t003:** The areas where the spot counts or rankings were analyzed.

Spot Location	Picture—Guide	Picture—Described (PD)	Methodology Used *
Left-side Nasal spots (LNS)—Used for PCA	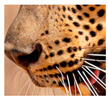	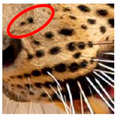	The LNS is the first row of spots in the left nasal area. The spot count in this area was carried out visually and recorded. The spot count was used for the analysis. These spots are commonly present in all leopards.
Left side Mystacial Row 1 (LM1)—Used for PCA	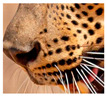	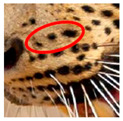	The cluster of spots immediately underneath the LNS. This cluster is within the mystacial area of the left side of the face. These spots are commonly present in all leopards. This cluster could be of varying patterns, and the patterns are not identical on both sides of the face. The spot count in this area was carried out visually and recorded. The spot count was used for analysis.
Left-side Mystacial 0Row 2 (LM2)—Used for PCA	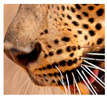	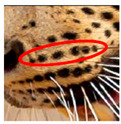	The row of spots immediately underneath LM1. This consists of a row of spots in all leopards. This row is within the mystacial area of the left side of the face. These spots are commonly present in all leopards. The spot count in this area was carried out visually and recorded. The spot count was used for the analysis.
Left-side Extra Spots Above (LAM2)—Used for PCA	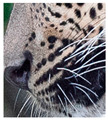	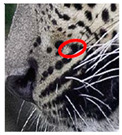	The spots that are positioned above LM2 but that are not a part of LM1. These spots are within the mystacial area of the left side of the face. These spots are not commonly present in all leopards. the spot count in this area was carried out visually and recorded. The spot count was used for the analysis.
Left-side Extra Spots Below (LBM2)—Used for PCA	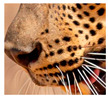	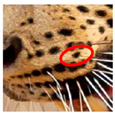	The spots that are positioned just below LM2. These spots are above the mystacial row below LM2. LBM2 consists of less spots than LM2. These spots are not commonly present in all leopards. These spots are within the mystacial area of the left side of the face. The spot count in this area was carried out visually and recorded. The spot count was used for the analysis.
Right-side Nasal spots (RNS)—Used for PCA	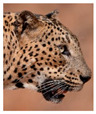	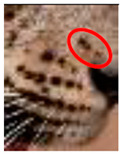	The RNS is the first row of spots in the right nasal area. The spot count in this area was carried out visually and recorded. The spot count was used for the analysis. These spots are commonly present in all leopards.
Right-side Mystacial Row 1 (RM1)—Used for PCA	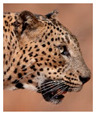	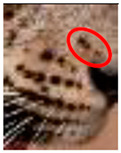	The cluster of spots immediately underneath the RNS. This cluster is within the mystacial area of the right side of the face. These spots are commonly present in all leopards. This cluster can be of varying patterns, and the patterns are not identical on both sides of the face. The spot count in this area was carried out visually and recorded. The spot count was used for the analysis.
Right-side Mystacial Row 2 (RM2)—Used for PCA	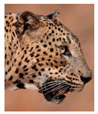	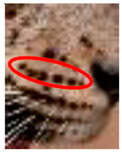	The row of spots immediately underneath the RM1. This consists of a row of spots in all leopards. This row is within the mystacial area of the right side of the face. These spots are commonly present in all leopards. The spot count in this area was carried out visually and recorded. The spot count was used for the analysis.
Right-side Extra Spots Above (RAM2)—Used for PCA	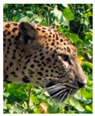	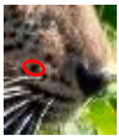	The spots that are positioned above RM2 but that are not a part of RM1. These spots are within the mystacial area of the right side of the face. These spots are not commonly present in all leopards. The spot count in this area was carried out visually and recorded. The spot count was used for the analysis.
Right-side Extra Spots Below (RBM2)—Used for PCA	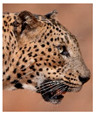	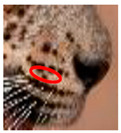	The spots that are positioned just below RM2. These spots are above the mystacial row, which is below RM2. RBM2 consists of less spots than RM2. These spots are not present in all leopards. The spot count in this area was carried out visually and recorded. The RBM2 spots were counted for the analysis.
The Guide Arch of the Forehead	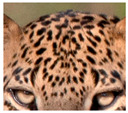	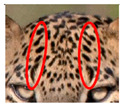	A prominent row of large spots starting from the top-inner corner of the area above each eye and that continues to the base of the ear lobe area. This row of spots forms a pattern similar with a less-angled arch and is present in all leopards. The formation of the pattern is not common to all leopards, as the spot sizes and shapes differ in each individual. We termed this pattern as the “guide arch”.
Elongated Spots in the guide arch of the Forehead	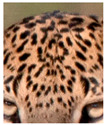	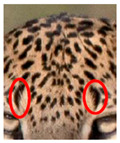	This is the first spot located in the guide arch area above the top-inner corner of each eye and has a comparatively longer shaped spot. We termed this spot as the “elongated spot”. Vibrissae is present in the periphery of the elongated spots on both sides of the forehead.
The genesis spots of the guide arch in the Forehead	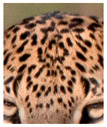	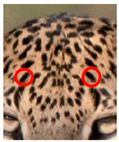	The spot directly above the elongated spot in the guide arch was termed as the “genesis spot”. This is the 2nd spot of the guide arch and is located on each side of the forehead.
Forehead 1 (F1)—Used for PCA	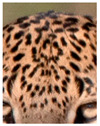	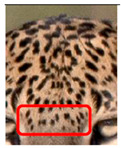	The spot count of the row of spots positioned between the lower portion of the elongated spot of the guide arches on either side of the face. The lower area of the elongated spot is the lower corner of the spot facing towards the top inner corner of the eye. The spots in the F1 area spread upwards, towards the spots of F2 and downwards towards the area between the eyes. The spot markings in this area are present in all leopards. The clearly visible spots in the area below F2 near the muzzle have all been counted as F1 spots.
Forehead 2 (F2)—Used for PCA	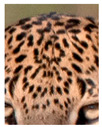	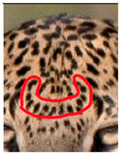	The F2 spots comprise the prominent formation that starts from the inner side of the left genesis spots of the left guide arch and continue to the inner side of the right genesis spot of the right guide arch. The number of spots in the F2 have been counted for the analysis. Any spots above the F2 spot for pattern formation are taken as F3 spots. Any spots below the F2 spot for pattern formation are taken as F1 spots.
Forehead (F3)—Used for PCA	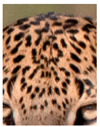	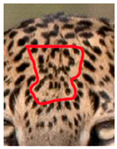	The F3 spots are the spots above and on the inner side of the F2 spot location for pattern formation. The F3 area continues until the 4th spot of the guide arch. The spots between the 4th spot of the left guide arch and the 4th spot of the right guide arch, both of which spread below F2 spot location for pattern formation were counted for the analysis.
Left side near eye (Leye)—Used for PCA	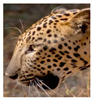	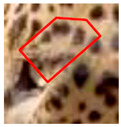	The number of spots in the semi-oval pattern starting above the inner area of the left eye and that continues to the outer end of the left eye. The number of spots was counted and used for the analysis. The pattern is present in all leopards. There are slight variations in the appearance of the pattern.
Side of the Face	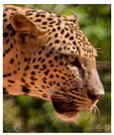	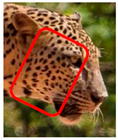	The side of the face is the area between the guide arch in the forehead, the mystacial area, and the neck area. This area only consists of spots of varying sizes and shapes. No rosettes were observed on the sides of the face of any leopard. Both sides of the facial area were considered for leopard identification.
Mandibular Spot Formation of both Sides of the face	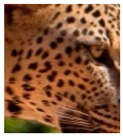	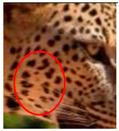	Distinctive pattern formation can be observed in the skin coat of the mandibular area on the side of the face of a leopard. A pattern of spots taking the shape of the circumference of an oblong or a circular pattern was present in all leopards observed by the team. The number of spots in the circumference of this pattern ranges between 9 and 13 spots. The inner side of the circumference of this pattern consists of a space with only between 1 to 3 spots and vibrissae in 1 or 2 points. This is the only area where vibrissae, apart from the mystacial area and the periphery of the elongated spots, is present in the side of the face. We termed this spot pattern as the “mandibular spot formation”.
The Upper Neck area (Both sides of the Leopard)	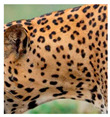	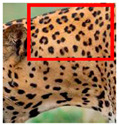	The upper neck area is the area between the side of the face and the foreleg. The upper neck is defined as the skin coat of the upper area between the rear upper corner of each earlobe and the front upper foreleg area. The upper neck area consists of rosettes and spots. Rosettes are more spaced out in the upper neck than in the other areas of the skin coat. This is present on both sides of the leopard.
Left-side Lower neck (LNeck)—Used for PCA	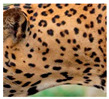	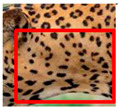	The number of broken lines consisting of spots starting from the left ear and ending in the lower neck area. The left foreleg area was visually observed and counted for the analysis. The pattern of broken lines consisting of spots is present in all leopards. The number of lines differ from each individual.
Right-side near eye (Reye)—Used for PCA	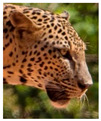	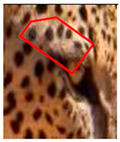	The number of spots in the semi-oval pattern starts above the inner area of the right eye and continues to the outer end of the right eye. The number of spots was counted and used for the analysis. The pattern is present in all leopards. There are slight variations in the appearance of the pattern.
Right-side Lower Neck (RNeck)—Used for PCA	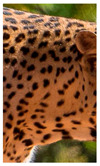	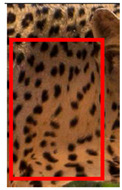	The number of broken lines consisting of spots, starting from the right ear and ending in the right neck area and the right foreleg area, was visually observed and counted for the analysis. The pattern of broken lines consisting of spots is present in all leopards, but the number of lines differs.
Left-side Upper Foreleg-Rank (LForeleg-Rank)—Used for PCA	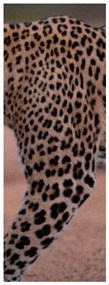	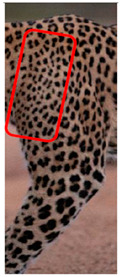	The spots formation along the left side of the upper foreleg was ranked considering the spacing of the spots. If the spacing between the spots is comparatively smaller, then the ranking is 2. If the spacing between the spots is higher, then the ranking is 3. This is a visual ranking based on the images collected.
Right-side Upper Foreleg-Rank (RForeleg-Rank)—Used for PCA	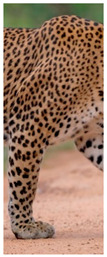	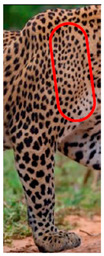	The spots formation of the right-side upper foreleg was ranked considering the spacing of the spots. If the spacing between the spots is comparatively smaller, the ranking is 2. If the spacing between the spots is higher, then the ranking is 3. This is a visual ranking based on the images collected.
Flank—Both sides of the Leopard	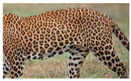	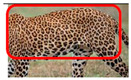	The flank of the leopard is described as the area between the tail and the lower foreleg area. The flank consists of spot and rosette formations in all leopards. The spot and rosette formations on both sides of the flank of the same leopard are different. The spot and rosette formations of each individual are different from each other. We did not come across any identical rosette formations in any of the leopards. We are continuing further studies on the spot and rosette formations on the flank.

* The described spot formation of the relevant location is marked in the PD column in red. The spot count within the marking was considered for the relevant field in the analysis in the mentioned locations.

**Table 4 animals-12-00660-t004:** The minimum population and density estimation of leopards at YNP 1.

Age Status	Female	Male	Minimum Population of Leopards (Combined Sexs)	Density of Leopards per Square Kilometer (km^2^) of the YNP1 (Area of YNP 1 Is 141 km^2^)
Listed as at the census date (31 March 2021)				
Sub Adult	05	07	12	
Early Adult	04	06	10	
Adult	16	13	29	
Aged Adult	07	02	09	
Total Listed Leopard at Census	32	28	60	
Unlisted but identified Leopards as at the Census date				
Cub	01	02	03	
Sub Adult	03	05	08	
Early Adult	Nil	Nil	Nil	
Adult	03	02	05	
Aged Adult	01	Nil	Nil	
Total Unlisted Leopards at Census	08	09	17	
Minimum Population of Leopards	40	37	77	0.5 (~1) Leopard per km^2^
**Sightings**	**Density/km^2^**	**% Co-efficient of variation**	**95%Cl/km^2^**	**Margin of error**
366	0.5461	53.04	19.25 ± 2.28	2.28

**Table 5 animals-12-00660-t005:** Probability of occurrence changes in the number of spots or in the patterns.

Spot Area	Obliterate Changes	Rejig Changes	Probability of Occurrence
Forehead 1	8	6	0.2373
Forehead 2	1	3	0.0677
Forehead 3	0	3	0.0508
Left Side Flank	0	1	0.0169
Left Side Mystacial Row 1	1	0	0.0169
Left side of the face	2	2	0.0677
Right Side Mystacial Row 1	0	1	0.0169
Right side Nasal Spots	1	0	0.0169
Right Side of the Face	1	0	0.0169

## Data Availability

Inquiries regarding the data can be directed to the corresponding author.

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
