# Peer review of "A Multi-Point Identification Approach for the Recognition of Individual Leopards (Panthera pardus kotiya)"

_animals, 2022, doi:10.3390/ani12050660_

Round 1

Reviewer 1 Report

This study is definitely worthy of publication. The multi-point approach is clearly superior to present methods. The oft-quoted assumption that leopards do not change their spots is also shown to be incorrect. However, the current MS has multiple flaws and much work will have to be done to revise it to a publishable standard.  In particular, the content needs to be completely revised so that it aligns with the conventional structure of a scientific paper. At present, ideas appear in inappropriate sections and often in more than one, leading to confusion and repetition. A number of detailed sections distract the reader from the central story, so should be presented in Supplementary Material for use by other felid biologists.

Change the title to ‘A multi-point identification approach…’  Multi-point should be hyphenated throughout.

The Simple Summary is too much like the abstract. It needs to be much shorter and simpler.

35 Insert comma after ‘time’.

47 Replace ‘Many’ with ’Much’.

48 Move ‘individual identification of leopards has been accomplished by using spot pattern markings or spot pattern variation’, to introduce leopards after the point about tigers. Insert scientific name for leopard. Change to something like ‘In the Felidae, the patterns displayed on the flank and face are highly variable.’

48 delete ‘family’. Ditto 54 and 95.

56 Delete ‘, and these differences’ and ‘of them’

60 Replace ‘gender’ with ‘sex’ here and throughout the MS. Sex is the correct biological term.

64 Delete full stop after ‘adopted’. Replace ‘where’ with ‘and’.

72 Delete ‘(Panthera tigris)’

76 Replace ‘were’ with ‘have been’.

78 Insert ‘The’ before ‘Capture’.   Delete ‘which was visibly’.

79 Replace ‘2’ with ‘two’

84 Delete ‘be known to’

87 Inconsistency in capitals for Capture-Recapture.  Delete’, which is practised internationally,’  Replace ‘These inherent weaknesses in the identification method will provide’ with ‘giving’

90 Delete ‘We observed them and photographically documented, during our research period, that the spot and rosette pattern formation in the skin of the leopards changed completely with injury. This change can result in the misidentification of a leopard, visually.’

94 Replace ‘For the current study,’ with ‘Our study species was’

96 Replace ‘They live’ with ‘It lives’

98 Contradiction: 95 says it was first described in 1816 by Lorenz Oken.

101 Include a reference to melanism.

103 This para presents the key findings of the study prematurely. At this stag it should indicate what the study set out to achieve.

108 Delete ‘widened our reseach and’

109 Delete ‘So far, the leopard identification was carried out by observing spots in a limited area.’

110 Insert ‘also’ before ‘observed’.  Delete ‘and observed that there is no interrelationship between the counts in a specific area in a leopard. In order, to identify an individual leopard we have to consider at least 8-10 points’.

  1. Replace ‘This’ with ‘Our aim was to develop a’. Replace ‘Panthera pardus kotiya’ with ‘the Sri Lankan leopard’.

114 Delete ‘could be used for the identification of individuals in other members of the cat family.’

Fig 1 is very low resolution.

127 Delete ‘observer travel distance was 100 km’s per day’. Replace ‘The travelling’ with ‘Travel’ and ‘were’ with ‘was’.

128 Replace ‘The observations of Leopard in each sighting was’ with ‘Leopards were’.

129 Replace ‘single-lens reflex camera with a digital imaging sensor (DSLR)’ with ‘single-lens reflex (DSLR) cameras’.

130 Delete ‘of such sightings’.

132 Replace ‘in Compact’ with ‘on compact’.

133 Move up ‘Telephoto and fixed lenses of various ranges were used in the cameras to capture the images.’ to follow sentence ending ‘cameras’ on 130, keeping the cameras and lenses together.

134 What does this sentence add?   Fig 2 mentions EXIF files, but no details are given in the text.

154 Nickname is one word.

163 Leopard should be lower case here and throughout the MS. Also, see earlier point about sex.

167 Lower case for coded. Replace ‘will remain’ with ‘is’.

168 What is the ‘codename’?

169 What exactly is a ‘record’, and how is it different from an ‘observation record’?

172 Replace ‘the’ with ‘a’ before census.

173 Which year?

175 I don’t understand the sentence beginning ‘A cross-sectional’ and I doubt if it adds anything so should be deleted.

179 What does ‘cage’ mean?

180 All lower case for ‘Identified Leopard Population’.

183 Delete ‘situation’.

  1. Change to ‘The Participants were selected by us and the DWC on a nonprobability expert stratified sampling method, based on their knowledge of the identified leopards (Supplementary Table 2). However, it is not clear what ‘a nonprobability expert stratified sampling method’ means, or how S2 contributes to this point.

185 I don’t understand the sentence beginning ‘The sample participant’.

187 Lower case for ‘Identification’. This publication should be cited as usual and listed in the References.

196 Lower case for ‘Categorised’.

Table 1. Lower case for ‘Adult’ and ‘Cub’.

203 Delete ‘The’.

205 Replace ‘and came to a conclusion on the Leopard's age’ with ‘to estimate age’.

206 I simply could not follow the point about unlisted leopards.

211 This crucial description of how individual leopards were identified should come before the section Coding of Identified Leopards.  It should also be expanded by extracting much of the descriptive details currently in Results.

Table 2. This large table should be in Supplementary Material. Lower case for all words: Spot, Left, Mystacial, Row, Right, Elongated, etc.

227 Delete the first sentence.

228 The second and third sentence belong in Discussion.

231 Delete the last sentence.

233 Change to simply ‘General markings’.

235 Delete ‘The black markings were termed as “Spots”’.

236 Replace ‘The spots are marked in figure 3A as A1’ with ‘(Figure 3A)’.

237 Delete ‘fur’. Replace (as shown in figure 3A as A2’ with ‘(Figure 3A)’.

237 Delete ‘skin’.

238 Lower case for more words: Spot, Rosette, Darker, etc.

258 Delete the last sentence.

282 Delete this unnecessary heading. This entire section is very detailed and specific, so would be better as Supplementary Material. Accordingly I haven’t made detailed editing suggestions, but it will need to be tightened up considerably. References to specific leopards (e.g. 336-338) should be deleted. 

285 Delete the second sentence.

449 This section also clearly belongs in Methods.

471 This section is appropriate in Results.

477 Delete the last sentence.

479 What does this para contribute? Delete.

489 Change ‘All’ to Together’.

490 Delete the sentence beginning “Since”.

513 Delete the second sentence.

514 Replace ‘This cluster’ with ‘Cluster1’.

516 The point beginning ‘This shows’ should be in Discussion.

522 Change to ‘in the mystacial, nasal, eye and foreleg areas’

525 Another comparison point for the Discussion.

537 This para should be in Methods. However, the last sentence is hardly needed.

542 The first sentence should be in the Intro.

513 Delete the last sentence.

560 Delete the second sentence.

561 Replace ‘Rejig’ with ‘Obliterate’.

565 Merge the last two sentences into one clear description of Rejig change.

577 The details of these six cases should be presented in supplementary material.

682 Ditto.

796 This section on sexing errors seems rather trivial and not obviously relevant to multi-point ID. Any such errors presumably would be soon corrected by later observations. If deemed useful, it should be relegated to supplementary material. Change ‘testicles’ to testes’.

The Discussion should be restructured. Much of it (e.g. para 1) simply repeats the results. A better approach would be to start with the ‘big’ ideas (e.g. 846), then highlight the problems and assumptions involved with natural markings, then provide your solutions. Finally, report the census results based on your improve method and explain the value of having the correct count.

Author Response

Response to Reviewer 1 Comments

This study is definitely worthy of publication. The multi-point approach is clearly superior to present methods. The oft-quoted assumption that leopards do not change their spots is also shown to be incorrect.

However, the current MS has multiple flaws and much work will have to be done to revise it to a publishable standard.  In particular, the content needs to be completely revised so that it aligns with the conventional structure of a scientific paper. At present, ideas appear in inappropriate sections and often in more than one, leading to confusion and repetition. A number of detailed sections distract the reader from the central story, so should be presented in Supplementary Material for use by other felid biologists.

Change the title to ‘A multi-point identification approach…’  Multi-point should be hyphenated throughout.

Corrected the title as the reviewer suggested.

The Simple Summary is too much like the abstract. It needs to be much shorter and simpler.

            The simple summary is corrected as suggested. 

35 Insert comma after ‘time’.

            Corrected as the reviewer suggested in line 35

47 Replace ‘Many’ with ’Much’.

Corrected as the reviewer suggested in line 48

48 Move ‘individual identification of leopards has been accomplished by using spot pattern markings or spot pattern variation’, to introduce leopards after the point about tigers. Insert scientific name for leopard. Change to something like ‘In the Felidae, the patterns displayed on the flank and face are highly variable.’

Corrected as the reviewer suggested

48 delete ‘family’. Ditto 54 and 95.

Corrected as the reviewer suggested

56 Delete ‘, and these differences’ and ‘of them’

Deleted as the reviewer suggested

60 Replace ‘gender’ with ‘sex’ here and throughout the MS. Sex is the correct biological term.

Corrected as the reviewer suggested in line 73,75.76

64 Delete full stop after ‘adopted’. Replace ‘where’ with ‘and’.

Deleted as the reviewer suggested

72 Delete ‘(Panthera tigris)’

Deleted as the reviewer suggested

76 Replace ‘were’ with ‘have been’.

Corrected as the reviewer suggested in line 76

78 Insert ‘The’ before ‘Capture’.   Delete ‘which was visibly’.

Corrected as the reviewer suggested

79 Replace ‘2’ with ‘two’

Corrected as the reviewer suggested

84 Delete ‘be known to’

Deleted as the reviewer suggested

87 Inconsistency in capitals for Capture-Recapture.  Delete’, which is practised internationally,’  Replace ‘These inherent weaknesses in the identification method will provide’ with ‘giving’

Consistency was maintained in capitals of Capture-Recapture.  Corrected as the reviewer suggested.

90 Delete ‘We observed them and photographically documented, during our research period, that the spot and rosette pattern formation in the skin of the leopards changed completely with injury. This change can result in the misidentification of a leopard, visually.’

Deleted as the reviewer suggested

94 Replace ‘For the current study,’ with ‘Our study species was’

Corrected as the reviewer suggested

96 Replace ‘They live’ with ‘It lives’

Corrected as the reviewer suggested

98 Contradiction: 95 says it was first described in 1816 by Lorenz Oken.

Agree with the reviewer. The Family Felidae was first described by Lorenz Oken. The idea given by the sentence was not clear. Therefore, removed the part ‘it was first described in 1816 by Lorenz Oken.’

101 Include a reference to melanism.

References included for melansim in line 165

103 This para presents the key findings of the study prematurely. At this stag it should indicate what the study set out to achieve.

            Corrected as suggested by the reviewer. The key findings were removed from the introduction part.

108 Delete ‘widened our reseach and’

Deleted as the reviewer suggested

109 Delete ‘So far, the leopard identification was carried out by observing spots in a limited area.’

Deleted as the reviewer suggested

110 Insert ‘also’ before ‘observed’.  Delete ‘and observed that there is no interrelationship between the counts in a specific area in a leopard. In order, to identify an individual leopard we have to consider at least 8-10 points’.

Corrected as the reviewer suggested

  1. Replace ‘This’ with ‘Our aim was to develop a’. Replace ‘Panthera pardus kotiya’ with ‘the Sri Lankan leopard’.

Corrected as the reviewer suggested

114 Delete ‘could be used for the identification of individuals in other members of the cat family.’

Deleted as reviewer suggested

Fig 1 is very low resolution.

The image of the study area was provided by the Department of Wildlife Conservation of Sri Lanka, who is the sole regulatory authority of Wildlife in Sri Lanka. The study area was also calculated by the Department of Wildlife Conservation of Sri Lanka. Due to the low resolution of the image, we have deleted it.

127 Delete ‘observer travel distance was 100 km’s per day’. Replace ‘The travelling’ with ‘Travel’ and ‘were’ with ‘was’.

Corrected as the reviewer suggested line 166

128 Replace ‘The observations of Leopard in each sighting was’ with ‘Leopards were’.

Corrected as the reviewer suggested

129 Replace ‘single-lens reflex camera with a digital imaging sensor (DSLR)’ with ‘single-lens reflex (DSLR) cameras’.

Corrected as the reviewer suggested

130 Delete ‘of such sightings’.

Deleted as the reviewer suggested

132 Replace ‘in Compact’ with ‘on compact’.

Corrected as the reviewer suggested

133 Move up ‘Telephoto and fixed lenses of various ranges were used in the cameras to capture the images.’ to follow sentence ending ‘cameras’ on 130, keeping the cameras and lenses together.

            Corrected as the reviewer suggested.

134 What does this sentence add?   Fig 2 mentions EXIF files, but no details are given in the text.

The word EXIF data means that the photo is stored in a manner that the original properties of the photo can be examined if necessary. The Word EXIF data is deleted from the figure 2 as it can be misleading. The sentence “stored in the same format as the original Photo” in the text implies that the original properties of the photo are available.

154 Nickname is one word.

Nickname was in one word format.

163 Leopard should be lower case here and throughout the MS. Also, see earlier point about sex.

Corrected all the inconsistencies in the word leopard as suggested. The term gender was replaced with sex.

167 Lower case for coded. Replace ‘will remain’ with ‘is’.

Corrected as the reviewer suggested

168 What is the ‘codename’?

It should be corrected as nickname.

169 What exactly is a ‘record’, and how is it different from an ‘observation record’?

It’s a mistake. We can delete the sentence “For each record”

172 Replace ‘the’ with ‘a’ before census.

Corrected as the reviewer suggested

173 Which year?

Year is 2021. The date is recorrected.

175 I don’t understand the sentence beginning ‘A cross-sectional’ and I doubt if it adds anything so should be deleted.

            We have deleted the word Cross sectional survey as it may complicate the understanding. We used cross sectional as it was conducted within a specific period of time, which was 15 Months.  The sentence will read as “a survey of sightings of identified individual Leopards was used to conduct the census”.

179 What does ‘cage’ mean?

            It is a box or an area  demarcated for the user to provide the required information in the census survey form

180 All lower case for ‘Identified Leopard Population’.

Corrected all the inconsistencies in capitalization of words.

183 Delete ‘situation’.

Deleted as reviewer suggested

183.Change to ‘The Participants were selected by us and the DWC on a nonprobability expert stratified sampling method, based on their knowledge of the identified leopards (Supplementary Table 2). However, it is not clear what ‘a nonprobability expert stratified sampling method’ means, or how S2 contributes to this point.

Non probability sampling is selecting the sample of participants to the census survey by the researchers. They are selected by judgement of the researchers and are not selected randomly. They were selected by basing their knowledge on the identified Leopards of the research area.

185 I don’t understand the sentence beginning ‘The sample participant’.

            .   It is a repetition of the earlier mentioned participants and could be deleted

187 Lower case for ‘Identification’. This publication should be cited as usual and listed in the References.

The capitalization of words was corrected. Entered the publication and listed in the references.

196 Lower case for ‘Categorised’.

The capitalization of words was corrected.

Table 1. Lower case for ‘Adult’ and ‘Cub’.

Capitalization of words were corrected.

203 Delete ‘The’.

Deleted as reviewer suggested

205 Replace ‘and came to a conclusion on the Leopard's age’ with ‘to estimate age’.

Corrected as the reviewer suggested

206 I simply could not follow the point about unlisted leopards.

Unlisted Leopards are those who were not qualified to be coded and Nicknamed as at the census date of 31/03/2021 due to them being below an estimated age of 1.5 years and those Leopards who are above 1.5 Years of age and were observed by us as after 31/03/2021, but during our research period and within the research area. It is explained in 221 and 222.

211 This crucial description of how individual leopards were identified should come before the section Coding of Identified Leopards.  It should also be expanded by extracting much of the descriptive details currently in Results.

Table 2. This large table should be in Supplementary Material. Lower case for all words: Spot, Left, Mystacial, Row, Right, Elongated, etc.

Currently, table 3 is not a description of how Leopards should be identified. It is the areas that we have used for spot counts or rankings that was used for data analysis. The identification descriptions are shown in the results. This description and the table have duplications of the same information, and as such, we have deleted the duplicated texts in the results and have amalgamated the data used for the analysis and the areas used for identification , in table 3.

227 Delete the first sentence.

Deleted as reviewer suggested

228 The second and third sentence belong in Discussion.

The second and third sentences moved to discussion as suggested.

231 Delete the last sentence.

Deleted as reviewer suggested

233 Change to simply ‘General markings’.

Corrected as reviewer suggested

235 Delete ‘The black markings were termed as “Spots”’.

Corrected as reviewer suggested

236 Replace ‘The spots are marked in figure 3A as A1’ with ‘(Figure 3A)’.

Corrected as reviewer suggested

237 Delete ‘fur’. Replace (as shown in figure 3A as A2’ with ‘(Figure 3A)’.

Corrected as reviewer suggested

237 Delete ‘skin’.

Corrected as reviewer suggested

238 Lower case for more words: Spot, Rosette, Darker, etc.

Capitalization of words were corrected.

258 Delete the last sentence.

Corrected as reviewer suggested

282 Delete this unnecessary heading. This entire section is very detailed and specific, so would be better as Supplementary Material. Accordingly I haven’t made detailed editing suggestions, but it will need to be tightened up considerably. References to specific leopards (e.g. 336-338) should be deleted. 

Corrected as the reviewer suggested and this section was moved to supplementary material.

285 Delete the second sentence.

Corrected as the reviewer suggested

449 This section also clearly belongs in Methods.

            The section moved to the methods as suggested.

471 This section is appropriate in Results.

This section was kept as the reviewer suggested

477 Delete the last sentence.

Deleted as the reviewer suggested

479 What does this para contribute? Delete

Deleted as reviewer suggested

489 Change ‘All’ to Together’.

Deleted as reviewer suggested

490 Delete the sentence beginning “Since”

Deleted as reviewer suggested

513 Delete the second sentence.

Deleted as reviewer suggested

514 Replace ‘This cluster’ with ‘Cluster1’.

Corrected as reviewer suggested

516 The point beginning ‘This shows’ should be in Discussion.

            The point was removed from the results and it is already in the discussion.

522 Change to ‘in the mystacial, nasal, eye and foreleg areas’

Corrected as reviewer suggested

525 Another comparison point for the Discussion.

            Included into the discussion as suggested.

537 This para should be in Methods. However, the last sentence is hardly needed.

            The paragraph included into the methodology

542 The first sentence should be in the Intro.

            The sentence was added into the introduction

513 Delete the last sentence.

Corrected as reviewer suggested

560 Delete the second sentence.

Corrected as reviewer suggested

561 Replace ‘Rejig’ with ‘Obliterate’.

Corrected as reviewer suggested

565 Merge the last two sentences into one clear description of Rejig change.

Corrected as reviewer suggested. The two sentences were merged as “The rejig change is when the spot or the rosette pattern changes its form and pattern from its originality in its prominence, shape, size and appearance.”

577 The details of these six cases should be presented in supplementary material.

Agree with the reviewer. This section was moved to supplementary material

682 Ditto.

Agree with the reviewer. This section was moved to supplementary material

796 This section on sexing errors seems rather trivial and not obviously relevant to multi-point ID. Any such errors presumably would be soon corrected by later observations. If deemed useful, it should be relegated to supplementary material. Change ‘testicles’ to testes’.

 Agree with the reviewer. This section was moved to supplementary material

The Discussion should be restructured. Much of it (e.g. para 1) simply repeats the results. A better approach would be to start with the ‘big’ ideas (e.g. 846), then highlight the problems and assumptions involved with natural markings, then provide your solutions. Finally, report the census results based on your improve method and explain the value of having the correct count.

            Discussion is restructured as suggested by the reviewer

Reviewer 2 Report

The study and methodology presented were adequate, the photographs demonstrate well the results discussed, I suggest that you use the data to propose an identification guidelines both for survey and population monitoring of the species as a parameter in other studies related to the animal for the quantification of examined animals. Congratulations.

Author Response

Response to Reviewer 2 Comments

The study and methodology presented were adequate, the photographs demonstrate well the results discussed, I suggest that you use the data to propose identification guidelines both for survey and population monitoring of the species as a parameter in other studies related to the animal for the quantification of examined animals. Congratulations.

Thank you for your kind response

Reviewer 3 Report

I think there could be merit to this paper that explains a new method for improving reliability in the identification of animals that have been remotely sampled. However, the paper currently is too long and very hard to follow. I find the results section particularly difficult to navigate. The paper needs to be rewritten and reduced in length so that only the clearest results that support the aims of the research are included.

The paper needs a thorough proof read to remove the errors in spelling and grammar. 

The methods, and particularly the data analysis section are not repeatable. It is essential this any published research be replicable to others (they can follow the published methods and run a similar study). Currently, there is insufficient details on how data have been analysed for this to happen. 

Simple summary

Quite technical and hard for the non-scientist or non-expert to understand. For example, methods such as capture-mark-recapture and the explanation of the leopard's coat pattern are very scientific. I recommend re-writing the simple summary to make it more accessible to the wider audience.

Abstract

I think the abstract is clear and provides some useful background information on why the research was run. Would it be possible to edit the introductory information and provide a brief explanation of the key results and a concluding sentence?

Introduction

Line 47: Much research has

Line 47: to recognise individual animals

Line 49: provide the scientific name the first time a species is mentioned in the text 

Line 50: of felids generally or leopards specifically?

Line 51: surely they are distinct to each individual? Why do you reach for genus?

Line 55: Please stick to the referencing convention of the journal. Tiger does not need a capital. 

Line 60: Unique to each individual animal.

Line 63/64: please check punctuation. Does Capture need a capital C?

Line 67: Is camera trapping a recent technology? It has been around for a long time now.

Line 72: You have already included the scientific name of the tiger earlier on, so you do not need to use it again. If you have a list of animals of the same Genus then you can abbreviate the Genus to the first letter, e.g. tiger (Panthera tigris), lion (P. leo), jaguar (P. onca) if they follow each other in a list and are not interrupted by other species with a different genus name. 

Line 74: I don't think you need common in the leopard's name if you are referring to the species level. You should include specific names if you are talking about individual subspecies, e.g. Amur leopard. The leopard is introduced earlier on in the introduction so remove the scientific name from here and include the first time you use this animal's common name.

Section 94 to 101 reads more like methods than introduction. 

Line 107: What is your hypothesis? How do you know the animal is injured if it is unobserved?

Line 113: Suggest you simply refer to the common name.

Methods

Figure 1 is of poor quality you cannot locate where the national park is.

The section on study area needs references.

Line 126: Data were obtained...

Line 128: Leopard does not need a capital

Figure 2: I think this is useful and provides the reader with a clear explanation of how data were sources. Please check the spelling, grammar and punctuation in all text boxes. No capital for leopard unless at the start of a sentence. No need for double spaces after a full stop. No space between full stop and the last letter of the sentence. 

Line 159: No need for a capital for morphological and suggest "from each leopard photograph". 

For the Coding and Data collection sections please have a read through and remove all unneeded capital letters and check sentence structure and flow. 

Line 210: except for leopards who do not...

Table 2: Is correlation the right words? To determine the identification of the same individual animal multiple times?

Data analysis

Please provide further information on how these tests were run. What data were included in each test? How were principal components extracted? 

Please explain the cluster analysis and the Euclidean distance. What do you mean by this and how/why is it important to your analyses?

Line 221: What data were pooled and for what analysis specifically?

The results section is very long and hard to follow. I am not sure what results I am looking for. There is a lot of detailed information on individual animals and the sets of photos, but I would recommend presenting results on the reliability of individual recognitions or the numbers of times that the methods accurately identified the same animal over time. The results currently seem to be more like an explanation of what data were reviewed. 

Table 4: I do not understand what this shows and the results that are presented in it. Please provide further explanation. 

Figure 5 is potentially useful but quite small and therefore hard to read. Please expand on the result shown here. 

Figure 6 is impossible to read. This needs to be clarified. 

The results section is far to long and the reader gets lost is trying to work out what is the key message from your data analysis. 

I would recommend a great deal of the information presented in the results be provided as supplementary information and simply distil down the most important results to support or refute your hypotheses in the actual results section.

Due to the weaknesses identified in the presentation of results, it is very difficult to understand the discussion and the key areas of evaluation and research extension.

The conclusion reads well and this is succinct. Can the basis of the results and discussion be formed around the clear points and take home messages of the conclusion? 

Line 933: Start with what you have found and then include why such ID methods need to be revisited later in the conclusion. 

Line 939: Best to not start a sentence with Because...

Author Response

Response to Reviewer 3 Comments

I think there could be merit to this paper that explains a new method for improving reliability in the identification of animals that have been remotely sampled. However, the paper currently is too long and very hard to follow. I find the results section particularly difficult to navigate. The paper needs to be rewritten and reduced in length so that only the clearest results that support the aims of the research are included.

The paper needs a thorough proof read to remove the errors in spelling and grammar. 

The methods, and particularly the data analysis section are not repeatable. It is essential this any published research be replicable to others (they can follow the published methods and run a similar study). Currently, there is insufficient details on how data have been analysed for this to happen. 

Simple summary

Quite technical and hard for the non-scientist or non-expert to understand. For example, methods such as capture-mark-recapture and the explanation of the leopard's coat pattern are very scientific. I recommend re-writing the simple summary to make it more accessible to the wider audience.

The simple summary was written again as suggested. 

Abstract

I think the abstract is clear and provides some useful background information on why the research was run. Would it be possible to edit the introductory information and provide a brief explanation of the key results and a concluding sentence?

The introductory section and concluding sentence were changed as suggested.

Introduction

Line 47: Much research has

Corrected as reviewer suggested

Line 47: to recognise individual animals

Corrected as the reviewer suggested

Line 49: provide the scientific name the first time a species is mentioned in the text 

Corrected as the reviewer suggested

Line 50: of felids generally or leopards specifically?

            It is of generally on felids because the same spot patterns and whisker spots could be identified in lions and jaguars as well.

Line 51: surely they are distinct to each individual? Why do you reach for genus?

            This could be applied to each and every individual within genus as well as within population.

Line 55: Please stick to the referencing convention of the journal. Tiger does not need a capital. 

Corrected as the reviewer suggested

Line 60: Unique to each individual animal.

Corrected as reviewer suggested

Line 63/64: please check punctuation. Does Capture need a capital C?

            Punctuations were corrected.

Line 67: Is camera trapping a recent technology? It has been around for a long time now.

            Agree with the reviewer. The sentence was changed accordingly.

Line 72: You have already included the scientific name of the tiger earlier on, so you do not need to use it again. If you have a list of animals of the same Genus then you can abbreviate the Genus to the first letter, e.g. tiger (Panthera tigris), lion (P. leo), jaguar (P. onca) if they follow each other in a list and are not interrupted by other species with a different genus name. 

            All the mistakes in the scientific names were corrected as suggested.

Line 74: I don't think you need common in the leopard's name if you are referring to the species level. You should include specific names if you are talking about individual subspecies, e.g. Amur leopard. The leopard is introduced earlier on in the introduction so remove the scientific name from here and include the first time you use this animal's common name.

            The common name was added as Sri Lankan leopard.

Section 94 to 101 reads more like methods than introduction. 

            The section was moved to methodology

Line 107: What is your hypothesis? How do you know the animal is injured if it is unobserved?

We have replaced the word Hypothesis with the word assumption. We have observed and documented some leopards before injury, With the fresh injury and monitored them after the injury. The spot and rosette patter formations have changed after injuries. However, documenting all injuries that occur of Leopards in the wild is not possible. But we have documented changes in spot and rosette formations in Leopards during continuous observations. We have assumed the cause of such changes to be due to injuries , as we have observed such changes after documented injuries of other Leopards.

Line 113: Suggest you simply refer to the common name.

            Replaced with the common name.

Methods

Figure 1 is of poor quality you cannot locate where the national park is.

The image of the study area was provided by the Department of Wildlife Conservation of Sri Lanka, who is the sole regulatory authority of Wildlife in Sri Lanka. The study area was also calculated by the Department of Wildlife Conservation of Sri Lanka. Due to the low resolution of the image, we have deleted it.

The section on study area needs references.

The reference was added.

Line 126: Data were obtained...

Corrected as suggested by the reviewer.

Line 128: Leopard does not need a capital

Corrected as suggested by the reviewer.

Figure 2: I think this is useful and provides the reader with a clear explanation of how data were sources. Please check the spelling, grammar and punctuation in all text boxes. No capital for leopard unless at the start of a sentence. No need for double spaces after a full stop. No space between full stop and the last letter of the sentence. 

            The mistakes were corrected as suggested

Line 159: No need for a capital for morphological and suggest "from each leopard photograph". 

Corrected as suggested by the reviewer.

For the Coding and Data collection sections please have a read through and remove all unneeded capital letters and check sentence structure and flow. 

Corrected as suggested by the reviewer.

Line 210: except for leopards who do not...

Corrected as suggested by the reviewer.

Table 2: Is correlation the right words? To determine the identification of the same individual animal multiple times?

            Agree. The correlation would not fit the sentence. The table legend was corrected as “The areas where the spot counts were analyzed to identify the individual leopards”

Data analysis

Please provide further information on how these tests were run. What data were included in each test? How were principal components extracted? 

This has explained in the results section

Please explain the cluster analysis and the Euclidean distance. What do you mean by this and how/why is it important to your analyses?

            Cluster analysis was done to demonstrate how the spot counts ranges between males and female and to check whether there is a significant difference among the male and female leopards.

Line 221: What data were pooled and for what analysis specifically?

            The spot data counted in both male and female at the 15 pointes were pooled and analyzed.

The results section is very long and hard to follow. I am not sure what results I am looking for. There is a lot of detailed information on individual animals and the sets of photos, but I would recommend presenting results on the reliability of individual recognitions or the numbers of times that the methods accurately identified the same animal over time. The results currently seem to be more like an explanation of what data were reviewed. 

Table 4: I do not understand what this shows and the results that are presented in it. Please provide further explanation. 

This table shows the gender-wise minimum population of identified Leopards in the study area as at 31/03/2021. This population was arrived by conducting a census survey of observations of Identified Leopards. The process of the survey is explained in lines 171 to 197.

Figure 5 is potentially useful but quite small and therefore hard to read. Please expand on the result shown here. 

             Figure 5 was enlarged

Figure 6 is impossible to read. This needs to be clarified. 

The image was enlarged and moved to supplementary images

The results section is far to long and the reader gets lost is trying to work out what is the key message from your data analysis. 

I would recommend a great deal of the information presented in the results be provided as supplementary information and simply distil down the most important results to support or refute your hypotheses in the actual results section.

            The results section was rearranged as instructed.

Due to the weaknesses identified in the presentation of results, it is very difficult to understand the discussion and the key areas of evaluation and research extension.

The conclusion reads well and this is succinct. Can the basis of the results and discussion be formed around the clear points and take home messages of the conclusion? 

            The conclusion was reformed as instructed,

Line 933: Start with what you have found and then include why such ID methods need to be revisited later in the conclusion. 

Our findings show that following an injury, spot numbers and spot or rosette patterns change. Since there is no association between spot counts in specific locations, multi-point identification must be used to identify an individual leopard or a new leopard. Because the spot patterns are likely to change over time, at least 9-10 characters should be examined to identify an exact individual.

Line 939: Best to not start a sentence with Because...

Round 2

Reviewer 3 Report

I am happy with the author's edits to this paper and feel that it is worthy for publication. I have three small suggestions to improve the quality of the paper overall.

  1. The section "Visual morphological description of a Sri Lankan leopard", I do not see how this is a result? I feel this should be moved into the method. It adds context to how data and information on each leopard was collected but it is not a result in itself. Results need to focus on the findings of the data collection.
  2. I recommend that the full explanation of the PCA be included in the data analysis section as this is what this section is for. It should fully explain how data were analysed, in what manner and in what order, and why the test was applied. The reader should fully understand how the testing was used before the results are explained. 
  3. Line 604, I recommend adding some context to this statement. Perhaps "Following the application of these described leopard survey methods, the population at this field site is estimated at..." I would change the sentence that starts with Because to "As we have shown that leopard spot patterns can change over time, we recommend..."

Author Response

Response to Reviewer 3 Comments

I am happy with the author's edits to this paper and feel that it is worthy for publication. I have three small suggestions to improve the quality of the paper overall.

2. The section "Visual morphological description of a Sri Lankan leopard", I do not see how this is a result? I feel this should be moved into the method. It adds context to how data and information on each leopard was collected but it is not a result in itself. Results need to focus on the findings of the data collection.

Agree with the reviewer. The visual morphological description moved into the Methodology section

3. I recommend that the full explanation of the PCA be included in the data analysis section as this is what this section is for. It should fully explain how data were analysed, in what manner and in what order, and why the test was applied. The reader should fully understand how the testing was used before the results are explained. 

Full explanation of PCA was included in the data analysis section in methadology.

  1. Line 604, I recommend adding some context to this statement. Perhaps "Following the application of these described leopard survey methods, the population at this field site is estimated at..." I would change the sentence that starts with Because to "As we have shown that leopard spot patterns can change over time, we recommend..."

Agree with the reviewer. The sentences were changed as the reviewer suggested
